# Direction for a Transition toward Smart Sustainable Cities Based on the Diagnosis of Smart City Plans

**Hee-Sun Choi [1] and Seul-Ki Song [2],\***

[1] Department of Planning and Strategy, Korea Environment Institution, Sejong 30147, Republic of Korea
[2] Division for Land Policy Assessment, Korea Environment Institution, Sejong 30147, Republic of Korea
\* Correspondence: sksong@kei.re.kr

**Abstract:** Achieving urban sustainability through smart cities is necessary to manage urban environmental problems that threaten human survival. Smart city policy emphasizes the environmental aspects of urban areas while embracing the social and economic sectors, allowing for the development of practical plans for urban sustainability. This study suggests smart sustainable city policy directions that can improve the transition to smart cities. It defines concepts such as smart sustainable cities, developing frameworks, and indicators. In this research, a smart sustainable city facilitated sustainable development by incorporating smart technologies into urban activities and services. In this study, indicators for smart sustainable city evaluation and diagnosis were derived. These were applied to selected case areas, such as Incheon Metropolitan City and Goyang-si, Gyeonggi-do in the Republic of Korea. These indicators play an important role in assisting policymakers in making decisions, simplifying a wide range of complex information and providing integrated perspectives on existing situations. The results of this study suggest transition directions for a smart sustainable city and application strategies for related plans and policies.

**Keywords:** smart city; sustainable city; smart urban plan; urban regeneration project; smart green city

## 1. Introduction

As 76% of the world's population is living in urban areas due to continued urbanization, securing the sustainability of cities is more important than ever [1–3]. Urban sustainability is the process of creating a pleasant urban environment for individuals, minimizing environmental problems caused by urbanization and creating cities that are safe and resilient to climate change and natural disasters [4,5]. The transition to sustainable cities is important, as cities with concentrated populations and resources play a central role in sustainable development and solving environmental problems [6,7]. To realize this, various concepts and types of sustainable urbanization have been proposed, including green, low-carbon, carbon neutral, ecological, and circular cities [8–13]. However, from the perspective of sustainable development, appropriate tools are required to manage cities, which are complex social, economic, and environmental systems. For this, several studies have emphasized technology and smart cities as alternative solutions [14,15]. The creation of smart cities has accelerated internationally with the advent of the Fourth Industrial Revolution. According to a survey by Markets and Markets, the global smart city market is expected to grow from $308 billion in 2018 to $617.2 billion in 2023 at an annual growth rate of 18.4% [16]. However, despite the declaration of urban sustainability, most smart city projects continue to have issues, such as increasing urban vulnerability and deepening personal information gaps due to uncertainty [17]. Thus, strategic approaches are required to promote smart cities as sustainable cities.

Therefore, this study suggests a policy direction toward a sustainable smart for a more progressive transformation of smart cities that are rapidly advancing as future cities. To this end, we present implications by deriving indicators that consider the "technology

application approach" and "demand-purpose-oriented approach", and applying them to smart city-related plans. Plan evaluations based on indicators present a desirable vision of the future and evaluate the plan's quality to see if it can be achieved, thus enabling an objective comparison of cities and serving as a guide in setting the direction and improving plans. In particular, by building an evaluation framework, this study demonstrates how urban sustainability can be achieved by proposing ideas for supplementing and improving smart city policies centered on applying smart technology to cities and elemental technology advancement, and improving complex urban problems. Ultimately, to explore the direction of the development of a smart city toward a sustainable city, we evaluate the existing plan by defining the concept of a "smart sustainable city" and developing diagnostic indicators, which may present the direction of the transition to a smart sustainable city and will be recommended for government policies and projects in the future.

This paper proceeds as follows. First, we review the literature on the concept and diagnosis of sustainable and smart cities. Second, we establish the concept of smart sustainable cities. Subsequently, after demonstrating the framework for urban development, a protocol is established to diagnose and evaluate smart sustainable cities, and evaluation indicators are derived accordingly. Finally, the level of each plan is evaluated by applying the derived diagnostic indicators to urban planning and smart city planning aimed at sustainable and smart cities. The procedure is illustrated in Figure 1.

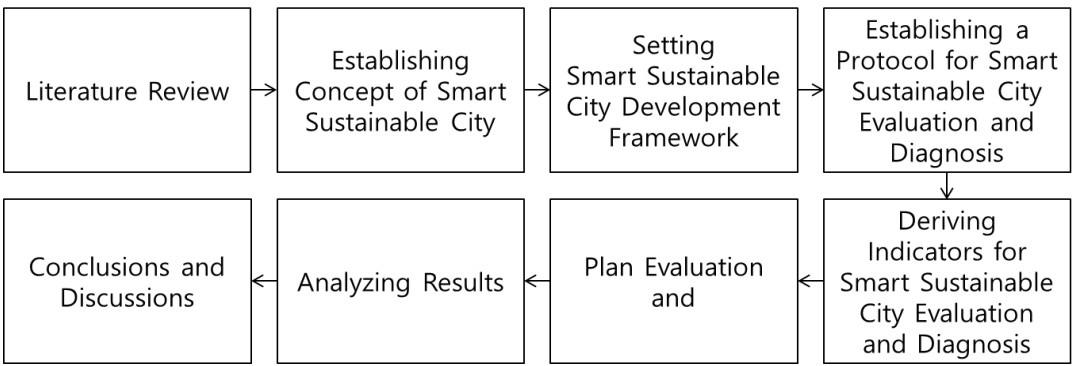

**Figure 1.** Research procedure.

## 2. Literature Review

### 2.1. Sustainable and Smart Cities

Since the 19th century, there have been attempts to focus on urban sustainability issues, although they did not attract much attention at the time [18]. However, as interest in environmental issues increased in the 1960s and 1970s, academics and planners began to recognize ecological approaches to urban planning and design. This led to the development of sustainability research and an increased interest in and understanding of the negative effects of urbanization on the ecosystem since the 2000s [19]. The ecological approach to urban planning provides a different perspective on sustainability from modernism, which was influenced by scientific rationalism and based on mechanistic and reductionist world-views, and post-modernism, which attempted to overcome the limitations of the scientific objectivity of modernism [19,20]. Figure 2 shows changes in the concept of sustainability with the transition from the modernist paradigm, which was based on economics, to ecological thinking. This conceptual difference exists because modernism focuses on technological and engineering infrastructure to plan a city as a separate component, and provides urban functions and a dualistic perspective that separates humans and the environment as two separate entities, while the ecological perspective regards the human and natural systems as dynamic, evolutionary, and interdependent [19,20].

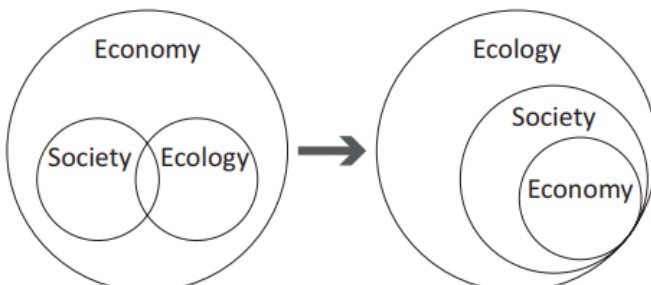

**Figure 2.** Changes in the concept of sustainability from the modernist paradigm to ecological thinking. Source: [20] (p. 56).

A new perspective on urban planning is needed to respond effectively to the vast changes that cities have undergone in the past decades, such as rapid urbanization, climate change, unsustainable energy systems, and socioeconomic crises. Many scholars see sustainable city development as a modern paradigm to solve these problems and create a desirable future of cities [21]. The concept of a sustainable city is a political initiative that responds to urban environmental problems caused during the 20th century, meaning "a city that can maintain natural resource supplies while achieving economic, physical, and social development and ensure protection from environmental risks which can potentially undermine the development" [18] (p. 1271). This is achieved while emphasizing strategies and processes for sustainability. Strategies and approaches to achieve urban sustainability differ between cities, as cities have unique aspects, including physical, climate, ecological, and economic, as well as unique demands from residents [18]. Transition plans to sustainable cities are a typical approach and reflect various factors, including energy conservation, the reduction of waste and pollutants, reduction of automobile traffics, conservation of open space and sensitive ecosystems, and conservation of favorable living and the cultural environment of local community, as key concepts. These plans consider density control, diversity, complex land use, compactness, sustainable transportation, passive solar design, and green or ecological design as key factors [11,22]. From this perspective, the most preferred urban models of sustainable cities, which proved to be a solution for various issues deteriorating urban sustainability over the past 20 years, are compact, eco, green, low-carbon, knowledge-based, resilient, and circulation city models [8–13]. In addition, smart cities have recently emerged as a new urban model that applies cutting-edge technology for urban operation and management and can efficiently respond to urban issues [8,11]. Smart city models are gaining attention, as the approach through smart systems and technologies has become more important than ever to solve challenges faced by future cities and ensure sustainability [6,23]. In terms of smart growth, "smart" city refers to a city that realizes sustainable development through cooperative decision-making processes and economic growth that does not destroy the environment [24]. Previously, the concept of smart growth was limited due to a lack of information and technology, but the Fourth Industrial Revolution made this possible [24]. According to McKinsey Global Institute [25], various smart city technologies contributed to enhance urban sustainability by reducing greenhouse gas emissions, water consumption, and waste.

*2.2. Definition and Diagnosis of Smart Cities*

Today, smart cities are a popular ideal, which originated from the belief that smart cities are managed and developed sustainably using information technology, engineering, infinite data, and energy resources; various related studies are conducted, and initiatives are established [26]. From an engineering perspective, the current concept of smart cities has evolved from digital, ubiquitous, and wired cities, suggesting that information and communications technology (ICT), knowledge, and environment are inextricably linked to the implementation of innovative cities [27–29].

Until now, the definition of a smart city varied depending on the economic level and policy of each country or region (city); there was no universally available definition, as

the concept emerged because of the needs of each country or city [30]. In the absence of a predetermined template and one-size-fits-all definition for the term "smart city", its definition encompasses various perspectives from ICT, education, and fair development to sustainability [31]. However, a common understanding is that various technologies help achieve sustainability in smart cities [32]. Toli and Murtagh [31] analyzed 43 studies to conclude that many definitions of smart cities deal with sustainability as their main goal. In summary, smart cities create knowledge and innovation; suggest an integrated and comprehensive vision of all aspects of city life, including economy, government, transportation, green, healthcare, and culture; and optimize the performance and efficiency of city processes, activities, and services by combining various components and actors in an intelligent system using ICT [28].

However, these urban ideals may appear differently in reality. A detailed examination of each definition of a smart city reveals that there are definitions that cover all three dimensions of sustainability (environmental, economic, and social aspects) and only one or two dimensions [31]. Many smart cities approached from an economic perspective often conflict with the environmental and social aspects of the city, thereby resulting in inequality from a social perspective due to the uneven distribution of the technological advantages of smart cities, or technological innovation or application that causes loss in terms of natural habitats and biodiversity [26]. Accordingly, Cugurullo [26] pointed out that smart urbanization has limitations in maintaining the balance of economy, society, and environment in terms of sustainability.

As the smart city was oriented in the definition, it is necessary to provide a direction for a desirable plan to move forward as a new model of a sustainable city. Various countries and organizations have already developed diverse evaluation systems, including standardization and certification systems in relation to urban models such as sustainable or resilient cities in terms of sustainability; many certification and standardization efforts have been used for smart cities [33–39]. However, unlike smart cities that claim to be sustainable, certification and standardization systems are still in their infancy, and focus on common technologies or detailed element technologies for smart homes, health, and transportation [40]; so, it is difficult to see that they fully incorporate sustainability.

In addition, the aforementioned smart city ideal cannot be achieved only by applying standardized technology, so considerations from various perspectives are required. For example, through city noise monitoring, optimization of the city noise problem could be promoted [41]; car-sharing business models could be improved by identifying accessibility problems and social exclusion through user opinion monitoring [42,43]. In addition, it is necessary to consider the impact of these technological applications on the health of users [44]. Issues still need to be discussed in implementation, such as information security issues regarding the use of personal information [45].

## 3. Method: Building a Framework and Deriving Indicators

Through the literature review, we found that existing smart cities and evaluation systems guiding them in a desirable direction does not sufficiently imply sustainability. Although various aspects of smart cities have value, policy direction and project execution for sustainable development and transformation require further consideration. This study examines how smart technologies can be properly introduced to increase the efficiency and effectiveness of urban planning, creation, and operation for the development of sustainable cities. This study seeks to improve the content and system of current plans by evaluating existing smart cities using the diagnostic indicators.

The derived diagnostic indicators were applied to a sustainable city and planning cases aimed at sustainable and smart cities to evaluate the level of each plan. In this study, smart sustainable city diagnostic indicators were applied to the Urban Master Plan, which is a comprehensive planning that suggests the basic spatial structure and long-term development direction of South Korean (hereafter Republic of Korea) cities, and acts as a foundation of the National Land Planning and Utilization Act in its jurisdiction (Special Metropolitan

City, a Metropolitan City, a Special Self-Governing City, a Special Self-Governing Province, or a Si/Gun [46]. Moreover, two smart city-related plans—Comprehensive Plan for Smart Cities and Urban Regeneration Revitalization Plan—are subjects for smart city-related plans for this study. First, the Comprehensive Plan for Smart Cities is a comprehensive plan that embodies the mid-to-short-term development direction of the city in the national plan [47]. It has the same jurisdiction unit as the Urban Master Plan but is established before the implementation of the smart city construction project and includes items related to the establishment, management, and operation of the smart city infrastructure and services [47]. Furthermore, the Smart Urban Regeneration Revitalization Plan is a plan for urban regeneration in accordance with the Special Act On Promotion Of and Support For Urban Regeneration, which incorporates smart urban infrastructure and services. In this case, the Urban Regeneration Revitalization Plan refers to a comprehensive action plan that links and comprehensively establishes various urban regeneration projects in the urban regeneration project region in accordance with the National Urban Regeneration Basic Policy and Urban Regeneration Strategy Plan [48]. Unlike the Urban Master Plan and the Comprehensive Plan for Smart Cities for the entire city, the Smart Urban Regeneration Revitalization Plan targets the community implementing urban regeneration projects.

### 3.1. Establishing the Concept of a Smart Sustainable City

According to the International Telecommunication Union (ITU) [49], which analyzed 116 smart city definitions, 26% were related to methods (ICT, information, communication, intelligence), 17% were related to purpose (environment, sustainability), and 17% were related to infrastructure and services [50]. Recently, the concept of "smart sustainable city" has been proposed to be complementary to smart cities [8,32,51–53]. ITU proposed the development of smart cities into smart sustainable cities for the development of their sustainability and defined smart sustainable cities as "innovative cities using ICT and other advanced technologies to improve citizens' quality of life and efficiency and competitiveness of city operation while meeting the needs of current and future generations in economic, social, environmental, and cultural aspects" ([52], p. 2). This study mainly adapted the ITU's definition of a smart sustainable city. The following definition, which developed the ITU's definition based on the advice and opinions of experts including the advisory group of this study, was used:

"Smart sustainable cities secure urban competitiveness in terms of environment, economy, society, and culture to meet the needs of the current and future generations, and aim for service innovation and efficiency by utilizing cutting-edge technologies for improving the quality of citizens' lives and urban ecosystems"

This definition includes:

- the beneficiaries of sustainable development: both current and future generations;
- the purpose of smart sustainable city: securing urban competitiveness and improving the quality of citizens' lives and urban ecosystem;
- the main means of urban development: advanced technology, including ICT; and
- the desired shape of a city: a city that continues to develop and aims for efficiency and innovation.

### 3.2. Smart Sustainable City Development Framework

Smart Cities Habitat Master Planning Framework [54], Smart Technology Engagement Framework for Smart Sustainable Cities [55], and the Garuda Smart City Framework (GSCF) [56] suggest varying composition and characteristics depending on the focus.

This study considers the smart sustainable city as the combination of smart technology, urban activities, and services. This combination enables cities to function and form a sustainable development path. Therefore, it was judged that smart technology enables the development of a sustainable city and creates benefits by improving the environmental, economic, and social aspects that frame sustainability. Based on previous research, the environmental, economic, and social aspects constituting sustainability are divided into eight

areas: natural environment, environmental impact and emissions, disaster, economy, social inclusion, basic services for quality of life, participation and cooperation, and governance. These eight areas are city components and must be managed responsibly by the city for current and future generations. Smart sustainable cities cannot be built in a short period of time but can be realized in the long-term through a process-oriented pathway resulting in feedback [24]. Technologies improve and mature over time, and smart enablers, such as smart technology, should design paths to increase the sustainability of the city through a series of decision-making processes. This means that eight areas of social, economic, and environmental aspects, such as the natural environment, should be continuously monitored based on smart technology and socioeconomic conditions. Based on this, indicators, policies, and systems in each field should be adjusted consistently toward sustainable development (Figure 3).

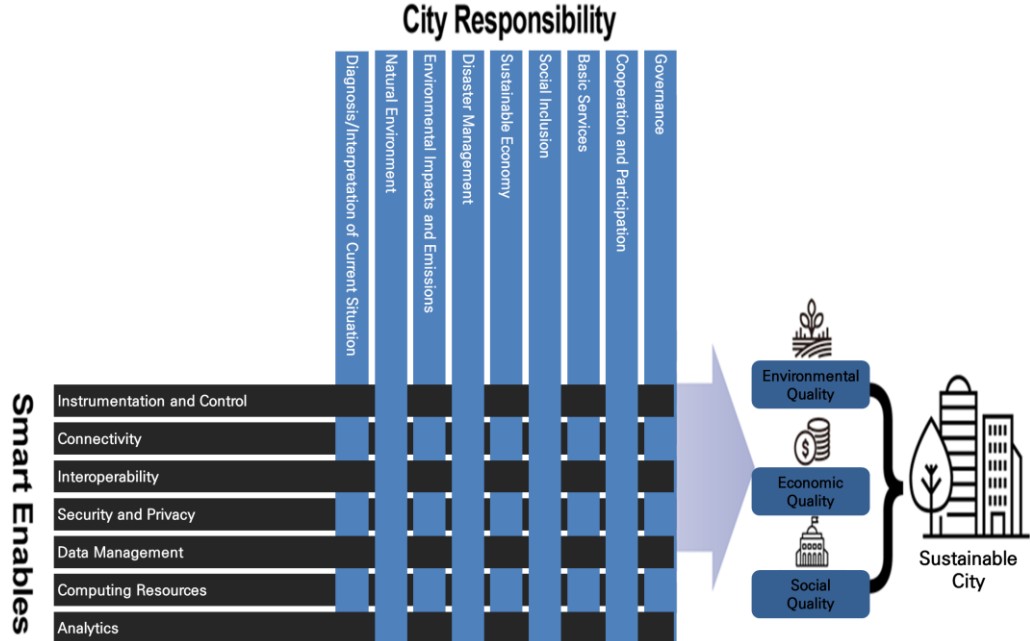

**Figure 3.** Smart sustainable city framework. Source: adopted from [57] based on [58].

*3.3. The Procedure of Deriving Indicators for Smart Sustainable City Evaluation and Diagnosis*

Smart sustainable city evaluation and diagnostic indicators should first set the items that constitute the plan-quality evaluation protocol for smart city planning. While it is difficult for planners and experts to agree on how "good" plans will be structured, there are several criteria for good plans shared among the quality assessment literature [59–62]. Although planning evaluation has various approaches depending on the discussion of planning theory, these protocols commonly focus on rational approaches that regard planning as a blueprint and define the quality of planning [63]. In this study, five components of the planning quality evaluation protocol were established based on the protocols of existing studies [59–62]:

1. Factual basis: Basic information for planning and implementation (current status and prospects);
2. Goals and objectives: Broad objectives to achieve the vision of the plan;
3. Policies and strategies: Specific, deliverable means and tools to implement vision and objectives;
4. Inter-governmental cooperation and public participation (cooperation and governance): Recognition of various stakeholders for the implementation of the plan and competence for cooperation between them;
5. Implementation: Specificity and possibilities for implementing objectives and policies.

### 3.4. Deriving Indicators for Smart Sustainable City Evaluation and Diagnosis

We created a framework and indicators to evaluate the sustainability and smartness of urban planning comprehensively. The indicators used to evaluate these five planning components were largely derived through two steps. First, the indicators (draft) were constructed based on previous studies and related evaluation indicators. Next, the final indicators were derived by adding, deleting, or supplementing through expert opinions based on the indicators (draft). Many countries and institutions have already developed smart city indicators considering sustainability and have evaluated and compared cities and used them for plans and policies. Various international standardization organizations, such as the International Standardization Organization (ISO) and ITU, develop standards related to smart city performance-evaluation indicators based on Sustainable Development Goal (SDG) 11 (sustainable city and community). Recently, the Republic of Korea began to prepare for the standardization of smart cities. Based on a previous study [53] that reviewed recent smart and sustainable city indicators in-depth, this study reviews seven international standards and SDG 11 developed by ISO, ITU, and European Standardization to evaluate the sustainability and smartness of cities. Two planning and strategic indicators that are broadly used are included in the review. For SDG 11, the national SDG indicator is also considered. In addition, as this study focuses on the Republic of Korea, smart city diagnostic indicators were based on the Korea Research Institute for Human Settlement (KRIHS) smart city diagnostic index, the Korea Institute of Civil Engineering and Building Technology (KICT) smart city evaluation index, and key basic principles in the ubiquitous urban planning of ACT in the construction, etc., of Ubiquitous Cities. The sustainable city index (Ministry of Environment urban regeneration guideline related to UN SDGs) is also reviewed (Appendix A, Table A2 [33–39,52,64–72]).

Considering the existing evaluation frameworks and methods related to smart sustainable cities reviewed earlier, the smart sustainable city evaluation and diagnosis indicators in this study are derived based on the following directions and principles. First, indicators (1) should be easily understood by stakeholders, (2) should be relevant to the interests of various stakeholders, (3) should be measured using available data at the city or country level, and (4) should be clearly related to the goals or possibilities of changes in urban policy [73]. In addition, indicators are classified into five categories according to the quality evaluation protocol of the plan derived earlier: factual basis, goals and objectives, policies and strategies, cooperation (governance), and implementation (Figure 4).

There are concerns that quantitative indicators such as the smart home ratio and smart city budget may fail to reflect regional characteristics and foster uniform plans. Therefore, we created a system that allows local experts to rate plans on a scale of 1–3 using content analysis. To avoid the establishment of a uniform plan that does not reflect regional characteristics, our indicators only present areas to be considered in planning for sustainability and smartness and some examples of technologies; essential technical elements are not mentioned in them. Experts who comprehensively considered the current status of the region, the plan's goals, and its vision were asked to judge whether the contents of the plan were appropriate and whether the goals could be achieved. Although being easy to use in a relatively simple manner, content analysis [74] lacked data reliability and objectivity; therefore, this study presented clear scoring criteria for evaluation reliability and objectivity. As such, the results would be the same if the same evaluators were to conduct the same evaluation again. In addition to increasing objectivity, we created a discussion process among experts during the evaluation process and adjusted their scores.

| Indicator Development Direction | • Balanced consideration of **urban sustainability and smartness:** 40 detailed indicators<br>• **Use of smart technologies to achieve** and smoothly implement **urban sustainability goals** |
|---|---|
| Indicator Development Principles | • Developing an indicator evaluation system that can reflect **regional particularities**<br>• **Evaluating the appropriateness of smart technology applications based on local conditions** |
| Diagnostic Indicator Protocol | 1. Factual Basis: basic current and prospective information for planning and implementation<br>2. Goals and Objectives: broad goals to achieve the plan vision<br>3. Policies and Strategies: specific and viable means and tools to implement the vision and objectives<br>4. Inter-Governmental/Organizational Cooperation and Citizen Participation: the cooperation of various stakeholders in plan promotion<br>5. Implementation: specific and actionable steps to implement goals and policies |
| Scale for each Diagnostic Indicator | • **0 points (none)**: The plan does not include any of the corresponding items<br>• **1 point (insufficient)**: The plan partially includes the items, but it is not specific and/or appropriate<br>• **2 points (medium-excellent)**: The plan fully includes the items, but it is not specific and/or appropriate<br>• **3 points (very excellent)**: The plan covers most items with specific content, and it is considers regional characteristics |

**Figure 4.** Principles and direction of smart sustainable city evaluation and diagnosis indicators. Source: [57].

First, the factual basis for the smart sustainable city evaluation and diagnosis indicators was largely divided into the analysis of the current status and future prospects and harmony with related plans from the sustainability and smart city aspects. Second, goals and objectives were evaluated after dividing them into goal setting based on current status analysis, connection and consistency with related plans, detailed goals, and measurement of goals from the sustainability and smart city aspects. Third, based on the sustainable city goals, policy and strategy assessed strategies for harmonizing with nature, and promoting disaster resilience, resilient and sustainable economies, social inclusion and cohesion, ability to provide equitable basic services, and improving quality. At this time, each detailed strategy was classified into sustainability and smart city aspects. Then, in cooperation (governance), the plan was evaluated by dividing the cooperation system, information disclosure and accessibility, and E-governance. Finally, the plan's implementation was evaluated by considering whether the policy was smart-based, the budget was efficiently executed, and the execution schedule was appropriately planned (Table 1).

**Table 1.** Smart sustainable city evaluation and diagnosis indicators.

| Main Category | Sub-Category | Indicator | Aspect | |
|---|---|---|---|---|
| 1. Factual basis | 1.1 Comprehensiveness of current status analysis | Level of comprehensive survey and consideration in various fields (land use, population, environment, transportation, finance, information and communication, smart infrastructure, etc.) | S | M |
| | | Appropriateness of application of smart techniques, such as big data and spatial information, in comprehensive analysis of current conditions | | M |
| | 1.2 Feasibility of future prospects | Appropriateness of future prospects, including population, economy, resource demand, and changes in urban industries and space following the Fourth Industrial Revolution | S | |
| | | Appropriateness of the use of big data, information, and prediction models for the advancement of future prospects | | M |
| | 1.3 Harmony with related plans | Consideration of urban and environment related higher-level plans | S | |
| | | Consideration of smart city related higher-level plans | | M |
| 2. Goals and objectives | 2.1 Objectivity of goal setting | Appropriateness of current status analysis and future prospects-based goal setting | S | M |
| | | Appropriateness of prospective of the goal and reflection of regional specialties | S | |
| | 2.2 Specificity of detailed objectives | Level of establishment of detailed goals for urban improvement based on smart technology reflecting the concept of sustainable development | S | M |
| | | Link and consistency with higher-level planning objectives and indicators | S | M |
| | 2.3 Measurement system of objectives | Appropriateness of setting target indicators and appropriateness of systematic and periodic evaluation system | S | |
| | | Appropriateness of information systems for periodic evaluation of goal execution and appropriateness of securing the system | | M |
| 3. Policies and Strategies | 3.1 Harmonization with nature and environmental impact | Water — Appropriateness of integrated water management plan (quality, aquatic ecosystem health, hydrology, sanitation, etc.) | S | |
| | | Water — Appropriateness of securing ICT solutions for water management such as water quality/hydrological warning system | | M |
| | | Air — Appropriateness of the plan considering reduction of air quality pollutant emissions and exposure | S | |
| | | Air — Appropriateness of securing the emission source management ICT system, such as real-time alarm and monitoring system | | M |
| | | Waste — Appropriateness of the plan for collection, treatment, and recycling of waste | S | |
| | | Waste — Appropriateness of the level of management efficiency through the provision of ICT-based waste collection-treatment-recycling solutions | | M |
| | | Carbon emission — Appropriateness of plan to reduce greenhouse gas and increase renewable energy usage | S | |
| | | Carbon emission — Appropriateness of ICT-based energy production-demand management and greenhouse gas emission management system | | M |
| | | Eco-system — Appropriateness of the plan to expand ecosystem services through the management of protected areas and urban ecosystem restoration. | S | |
| | | Eco-system — Appropriateness of ICT-based ecosystem monitoring and management system | | M |
| | | Noise — Measures for reducing noise level for the areas exceeding noise exposure limits | S | |
| | | Noise — Appropriateness of ICT-based noise level monitoring | | M |
| | | Environmental awareness — Level of consideration for online/offline education and policies to improve environmental awareness | S | M |
| | | Environmental awareness — Appropriateness of smart environmental education and nudge policy for environmental awareness improvement | | M |
| | | Land use — Detail of green infrastructure expansion plan (green transformation of infrastructure), such as respecting natural topography | S | |
| | | Land use — Appropriateness of ICT-based land-use change, damage, and urban ecosystem management | | M |
| | 3.2 Promoting disaster resilience | Disaster damage control — Appropriateness of disaster prevention and damage control plans based on analysis of disaster vulnerability in dangerous areas | S | |
| | | Disaster damage control — Vulnerability analysis and plan using big data analysis, etc. | | M |
| | | Customized integrated urban disaster crisis management — The level of securing the disaster response and crisis management system, such as protection of vulnerable groups | S | |
| | | Customized integrated urban disaster crisis management — Appropriateness of an efficient step-by-step disaster management system using ICT (alarm/evacuation/emergency rescue/recovery system, etc.) | | M |

**Table 1.** Cont.

| Main Category | Sub-Category | | Indicator | Aspect | |
|---|---|---|---|---|---|
| | 3.3 Resilient and sustainable economy | Economic sustainability | A plan that takes the economic sustainability of the lower income bracket into account(vocational education programs, social jobs, community jobs, etc.) | S | |
| | | | Level of community-based economic revitalization plan (business support, investment, etc.) compatible with the vision, characteristics, and cultural values of communities, as well as the projects that meet the needs of the region | S | |
| | | | Appropriateness of promoting innovation in smart city services (creative enterprises, startups, ICT R&D investment, etc.) and ICT-based green industries and jobs | S | M |
| | | Economic resilience | The level of consideration for economic recovery plans based on external factors such as disasters | S | |
| | | | Appropriateness of minimizing economic activity restrictions due to disaster using smart technology | | M |
| | 3.4 Social inclusion and cohesion | Improvement of the living environment | Identification and improvement of buildings vulnerable to disasters and poor housing (residential environment of vulnerable class) | S | |
| | | | Appropriateness of expansion of green building and green community | S | M |
| | | | Appropriateness of smart technology application for improvement of (aged) residential environment | | M |
| | | Social cohesion | Level of service for community activation | S | |
| | | | Appropriateness of public space planning and design considering the historical nature and culture of the community | S | |
| | | | Level of securing a smart plan foundation for social cohesion, such as smart living lab and smart neighborhood meeting | | M |
| | 3.5 Providing reasonable basic services and improving quality | Water supply | Appropriateness of supply-demand management plan for sustainable water supply | S | |
| | | | Water monitoring using ICT (demand supply: application of water meters, etc.) and technology introduction | | M |
| | | Power supply | Appropriateness of a systematic management plan for a sustainable electricity system, such as production-supply-demand management | S | |
| | | | Appropriateness of technical use for power supply and demand management, such as Smart grid and Electrical meter | | M |
| | | Transportation | Level of securing public transportation infrastructure considering the mobility of citizens, including vulnerable groups (children, persons with disabilities, senior citizens, etc.) | S | |
| | | | Appropriateness of smart green traffic planning (Intelligent Integrated Traffic Monitoring and Management System, Multimodal Service, Smart Public Bicycle Sharing System, etc.) | | M |
| | | Medical service | Level of securing equity and fairness in access to quality health services | S | |
| | | | Appropriateness of the application of ICT-based customized medical assistance measures for basic health care (remote medical treatment, electronic health and medical records, etc.) | | M |
| | | Education | Appropriateness of lifelong education service plan considering equity in access | S | |
| | | | Appropriateness of smart systems (network, device, program level, etc.) to expand access to education | | M |
| | | Public safety | Appropriateness of safety plan for accidents, crimes, fires, etc. | S | |
| | | | Appropriateness of the application of ICT systems to ensure public safety (digital surveillance cameras, etc.) | | M |
| | | Accessibility to green space | Diversity of open parks and green areas and accessibility for all classes | S | |
| | | | Appropriateness of smart technology application for the use and management of parks and green areas | | M |
| | | Public facilities/space accessibility | Level of equitable distribution of public facilities and securing accessibility for all classes | S | |
| | | | Appropriateness of application of smart facility management technology to improve visitor utilization and accessibility | | M |
| | | History, culture, leisure | Appropriateness of the plan to preserve historical and cultural resources and expand cultural and leisure opportunities | S | |
| | | | Level of securing accessibility, such as online reservation of public recreation services | | M |

**Table 1.** Cont.

| Main Category | Sub-Category | Indicator | Aspect | |
|---|---|---|---|---|
| 4. Cooperation (governance) | 4.1 Cooperation system | Gain a foundation for cooperation between various departments and ministries | S | |
| | | The role and participation level of various stakeholders, such as vulnerable groups and experts, etc. | S | |
| | 4.2 Information disclosure and accessibility | Provide an easy-to-understand form of information for all participants | S | |
| | | Utilization of various communication channels including ICT for information transmission | | M |
| | 4.3 E-governance | The level of infrastructure (roadmap, Internet network, etc.) and platform construction for securing the ease of participation of citizens based on ICT (anonymous feedback mechanism, etc.) | | M |
| | | Appropriateness of organization's internal work process digitization plan and establishment of foundation (connection technology, competency, etc.) | | M |
| 5. Implementation | 5.1 Policy support system | Secure smart service reliability system (Information security) | | M |
| | | Securing changeability of regulatory framework for ease of ICT usage (regulatory sandbox) | | M |
| | 5.2 Budget | Appropriateness of funding plan proposals for sustainability strategies | S | |
| | | Appropriateness of the proposal of a financing plan for smart city strategy | | M |
| | 5.3 Schedule | Establishing a clear time span, such as the implementation stage of the plan: Sustainability | S | |
| | | Establishing a clear time span, such as the implementation stage of the plan: Smart city | | M |
| Total | Total five categories | 73 subcategories | 40 | 40 |

Note: S: Sustainability, M: Smartness.

## 4. Case Study Analysis and Results

### 4.1. Target Location and Plan

This study sought to apply diagnostic indicators to plans for smart cities or plans highly relevant to smart cities. Although the purpose of the spatial hierarchy and plan is different, the evaluation was conducted on urban master plans, comprehensive plans for smart cities, and smart urban regeneration revitalization plans, which deal with the concept of a smart city. Two case cities in Republic of Korea, where all three plans were established, including one metropolitan city and one city, were reviewed. The urban master plan is a legal plan based on the National Land Planning and Utilization Act and mandated to be established by all jurisdictions with a population of 100,000 or more [46]. Comprehensive plans for smart cities are required to be established before the project is implemented [47]. Thus, smart city plans were established only in some cities, as not all municipal governments are obliged to establish one. In addition, seven locations (Pohang, Gyeongbuk, etc.) completed smart urban regeneration revitalization plans and five locations (Buk-gu, Daegu, etc.) were under planning [16]. Among local governments with comprehensive plans for smart cities, locations where smart urban regeneration revitalization plans were established were reviewed. Incheon Metropolitan City and Goyang-si, Gyeonggi-do were selected as target areas. The previously derived Smart sustainable city evaluation and diagnosis indicators (Table 1) were applied (Figure 5).

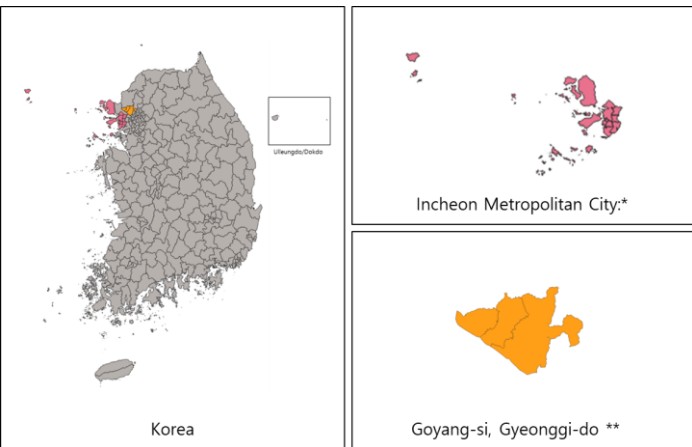

**Figure 5.** Plan evaluation target site. Notes: * Metropolitan City. ** City (Si) (with urban form and a population of more than 50,000).

### 4.2. Evaluation Method

Urban master plans, comprehensive plans for smart cities, and smart urban regeneration revitalization plans in Incheon Metropolitan City and Goyang-si, Gyeonggi-do were diagnosed and evaluated by 10 experts including local experts [75–80]. In order to increase the objectivity of the score, the score was adjusted through discussion among experts.

### 4.3. Regional Analysis Results

Figure 6 shows the scores of the six plans in sustainability and smartness criteria. The overall sustainability and smartness scores were expressed by converting the sum of the average scores (up to 3 points each) for each classification of sustainability and smart indicators (five categories, up to 15 points) into a total of 100 points.

First, the average sustainability score of the six plans was 45.58 points, less than half, and the lowest score was 37.80 points (Bupyeong, Incheon Metropolitan City Smart Urban Regeneration Revitalization Plan). The highest score was 52.74 points (Hwajeon-dong, Goyang-si Smart Urban Regeneration Revitalization Plan). There was no significant difference in scores in the four local government plans, except for the Smart Urban Regeneration Revitalization Plan, which is a community-level project.

The average score of smartness was 23.14 points, which was very low compared to the sustainability scores, and it was relatively lower in four plans except for the Comprehensive Plan for Smart Cities (Goyang-si: 38.46 points, Incheon Metropolitan City: 44.88 points). The lowest was 11.05 points (Goyang-si Smart Urban Regeneration Revitalization Plan), and the maximum was 16.98 points (Goyang-si Urban Master Plan).

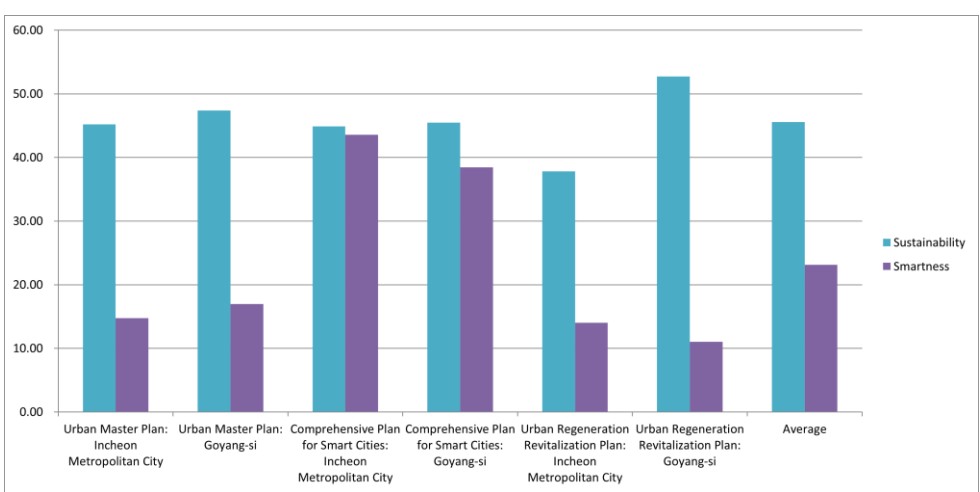

**Figure 6.** Overall score for sustainability and smartness by plan.

Despite the differences in the size of the city, Incheon Metropolitan City and Goyang-si did not have a large score difference in the Urban Master Plan and Comprehensive Plan for Smart Cities; however, Goyang-si received higher scores in the Smart Urban Regeneration Revitalization Plan at the community level. This was likely because the process of establishing the Urban Master Plan and Comprehensive Plan for Smart Cities in Republic of Korea had a uniform and formal structure rather than considering the characteristics of the city. Conversely, it was found that the Smart Urban Regeneration Revitalization Plan established at the community level could have a large score difference depending on the interest and ability of the local residents.

Figure 7 and Table 2 are the result of dividing the scores of each major classification by each plan into sustainability and smartness. In terms of sustainability, the Urban Master Plan received relatively low scores in the cooperation and governance criteria, and overall, the smartness side received low scores compared to sustainability. The Comprehensive Plan for Smart Cities shows relatively similar patterns in sustainability and smartness. Unlike the Urban Master Plans and Comprehensive Plans for Smart Cities, which comprehensively deal with various urban elements, the Smart Urban Regeneration Revitalization Plan, which is a community-level project, scored relatively high in cooperation, governance, and implementation in terms of sustainability, but scored low in terms of factual basis, goals and objectives, and policies and strategies. In terms of smartness, overall, it received a low score (not specified).

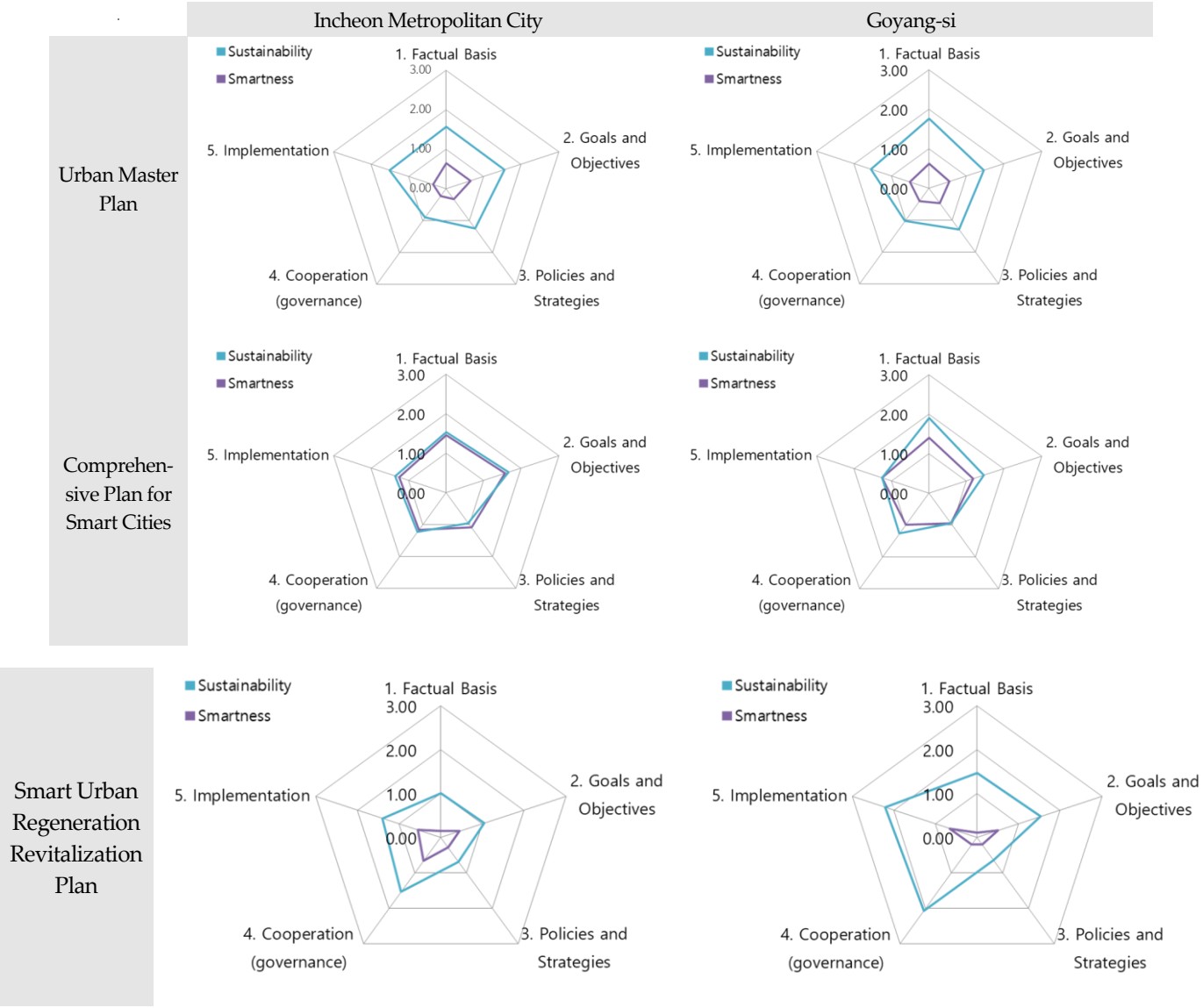

**Figure 7.** Sustainability and smartness score (main category) via plan.

**Table 2.** Sustainability and smartness score (sub-category) via plan.

| Category | | Incheon Metropolitan City | | | | | | Goyang-si | | | | | |
|---|---|---|---|---|---|---|---|---|---|---|---|---|---|
| | | A | | B | | C | | A | | B | | C | |
| | | S | M | S | M | S | M | S | M | S | M | S | M |
| **1** | 1.1 | **1.9** | 0.9 | **1.9** | **1.6** | 1.2 | 0.2 | **2.0** | 0.8 | **2.1** | 1.4 | **1.8** | 0.2 |
| | 1.2 | **1.5** | 0.4 | 1.2 | 0.8 | 1.0 | 0.2 | 1.4 | 0.5 | **1.6** | 1.0 | 0.6 | 0.0 |
| | 1.3 | 1.3 | 0.4 | **1.5** | **1.9** | 0.8 | 0.0 | **1.9** | 0.4 | **2.0** | 1.9 | **2.0** | 0.0 |

**Table 2.** Sustainability and smartness score (sub-category) via plan.

| Category | | Incheon Metropolitan City | | | | | | Goyang-si | | | | | |
|---|---|---|---|---|---|---|---|---|---|---|---|---|---|
| | | A | | B | | C | | A | | B | | C | |
| | | S | M | S | M | S | M | S | M | S | M | S | M |
| 2 | 2.1 | **2.0** | 1.1 | **1.9** | **2.0** | **1.6** | 0.4 | **2.0** | 1.1 | **1.9** | **1.7** | **1.9** | 0.4 |
| | 2.2 | 1.3 | 0.5 | **1.7** | **1.6** | 0.6 | 0.5 | 1.1 | 0.4 | 1.3 | 1.1 | 1.1 | 0.6 |
| | 2.3 | 1.3 | 0.5 | 1.3 | 1.1 | 0.8 | 0.4 | 1.1 | 0.3 | 1.0 | 0.8 | 1.0 | 0.4 |
| 3 | 3.1 | 1.1 | 0.3 | 0.8 | 0.8 | 0.3 | 0.1 | 1.3 | 0.3 | 0.8 | 0.7 | 0.3 | 0.0 |
| | 3.2 | **1.6** | 0.4 | 1.3 | 1.4 | 0.4 | 0.1 | **1.8** | 0.7 | 1.2 | 1.2 | 0.4 | 0.3 |
| | 3.3 | 0.9 | 0.3 | 1.0 | 1.1 | 1.2 | 0.8 | 1.0 | 0.6 | 0.8 | 0.7 | 1.3 | 0.4 |
| | 3.4 | 1.2 | 0.3 | 1.0 | 1.3 | 1.2 | 0.5 | 1.1 | 0.4 | 1.0 | 1.1 | **1.5** | 0.7 |
| | 3.5 | **1.5** | 0.4 | 1.0 | 1.2 | 0.7 | 0.3 | **1.5** | 0.6 | 1.1 | 1.2 | 0.7 | 0.2 |
| 4 | 4.1 | 0.9 | | 1.4 | | **1.7** | | 1.1 | | **1.5** | | **2.4** | |
| | 4.2 | 0.9 | 0.2 | 0.9 | 1.1 | 1.2 | 1.0 | 0.9 | 0.5 | 0.9 | 0.8 | 1.4 | 0.2 |
| | 4.3 | | 0.3 | | 1.2 | | 0.5 | | 0.4 | | 1.1 | | 0.2 |
| 5 | 5.1 | | 0.2 | | 0.9 | | 0.3 | | 0.3 | | 1.1 | | 0.2 |
| | 5.2 | 1.4 | 0.4 | 1.2 | **1.5** | 1.4 | 0.6 | **1.7** | 0.5 | 1.2 | 1.4 | **2.2** | 1.2 |
| | 5.3 | **1.6** | 0.6 | **1.5** | **1.7** | 1.4 | 1.0 | 1.4 | 0.9 | 1.3 | **1.5** | **2.2** | 1.0 |

Note: A = Urban Master Plan, B = Comprehensive Plan for Smart Cities, C = Smart Urban Regeneration Revitalization Plan, S = Sustainability, M = Smartness, Blanks (Grey) = No indicator, Bold = over 1.5 points, Category = see Table 1.

All three plans require overall improvement in both sustainability and smartness. Table 3 shows the issues to be considered in establishing future plans for each item through analysis of scores for each detailed sector.

**Table 3.** Considerations for planning.

| Category | Plan | Implications |
|---|---|---|
| Factual basis | A | Some degree of sustainability is considered, but not properly analyzed or not reflecting specific or regional characteristics (1 to mid-2 points). However, the current status analysis is the foundation of the goals and strategies of the plan, so it needs to be more specific. Further, analysis of regional issues and potentials, and clear understanding of future prospects is required. To this end, it is desirable to conduct an analysis using smart technology (big data, etc.), but there is a lack of appropriate application of smart technology in planning. |
| | B | Compared with the urban master plan, sustainability received similar scores, and smart scores were somewhat higher. Given that the plan focuses on smartness, the smart scores received are not high enough (less than 2 points). The smart urban plan needs to consider sustainability in the current status analysis and reflect adoption of smart technology in future plans. |
| | C | Current status analysis and future prospects are not achieved in various aspects owing to the nature of the short-term project, and the analysis using smart technology is limited. The current status analysis is the foundation for the goals and strategies of subsequent projects. Therefore, it is necessary to find smart technologies that can be used for the current status analysis considering the characteristics of urban revitalization projects, such as collecting residents' opinions through smart living lab, building regional platforms, and data management. |

**Table 3.** Considerations for planning.

| Category | Plan | Implications |
|---|---|---|
| Goals and objectives | A | Outlined in terms of sustainability, but not specific or appropriate (high-1 point–2 points). Compared to sustainability in terms of smartness, the goal setting, detailed milestone setting, and target measurement system are not considered.In particular, the target measurement system is associated with the feedback system of the future implementation, monitoring, and improvement direction plan, and it needs to be supplemented. |
| | B | Sustainability and smartness scores were at similar levels (mid-1–2). As with the basic urban plan, improvement is required because the goal setting, detailed milestone setting, and target measurement system are not fully considered. |
| | C | The urban revitalization plan presents specific goals, such as residential improvement and revitalization of the local economy, since the main purpose of the project is urban revitalization. However, considering that these plans aim to promote smart urban revitalization, it is necessary to differentiate them from other types of urban revitalization projects in goal settings related to smartness. |
| Policies and Strategies | A | All three plans focus on physical improvement and supply of infrastructure but do not adequately reflect aspects such as harmony with nature, social equity, inclusion, and resilience. Strategies related to smartness are rarely considered in basic urban and urban revitalization plans. The smart urban plan reflects the smart aspect, but it is difficult to say that it is applied considering sustainability because it only applies smart technology that was introduced from the U-city policy rather than as a means to achieve sustainability. The direction of sustainability and measures to reflect various smart strategies in the plan as a means of sustainability are required. |
| | B | |
| | C | |
| Cooperation (governance) | A | The most inadequate sector alongside the strategies in both basic urban and smart urban plans (1 point range: partially mentioned but not specific or appropriate). The urban revitalization plan receives relatively high scores in establishing a cooperative system given the nature of urban revitalization projects based on residents' participation. All three plans score close to zero in smartness, indicating that they are rarely considered in the plan. This is due to the lack of smart technologies or planning techniques to promote cooperation systems as well as lack of practical considerations despite the importance of cooperation and governance in planning, so additional related smart technologies and planning techniques are required. |
| | B | |
| | C | |
| Implementation | A | Smartness was hardly reflected in implementation. |
| | B | Among the detailed areas of implementation, the policy support system scored relatively low and needs to be improved. |
| | C | Although it received relatively high scores in implementation compared to the other two plans in terms of sustainability, it is necessary to discover smart technologies and planning techniques that can be used in the implementation due to low scores in terms of smartness. |

Note: A = Urban Master Plan, B = Comprehensive Plan for Smart Cities, C = Smart Urban Regeneration Revitalization Plan.

## 5. Conclusions and Discussions

### 5.1. Conclusions

Despite the various advantages of smart cities, various factors need to be considered in terms of the sustainable development of cities. This study was conducted to suggest improved policy directions in terms of transition to allow cities to secure sustainability and smartness. This study sought a system to diagnose the plan for transition to a smart sustainable city by establishing a protocol that generally constitutes the plan and deriving indicators that constitute the sustainability and smartness of the plan at each stage.

The diagnosis and evaluation plan evaluated the Urban Master Plan, which is the highest-level plan at the city level, Comprehensive Plan for Smart Cities, which includes the smart urban plan direction and major tasks, and Smart Urban Regeneration Revitalization Plan, which involves smart-city-type revitalization for existing urban space. Through the evaluation, this study explored the level, limitations, and future development of Republic of Korea's major plans in terms of smart sustainability. The smart sustainable city evaluation and diagnostic indicators presented in this study are expected to be a basic consideration for the future planning of urban development by policymakers. Relevant factors should be taken into consideration for urban development. However, as the protocols and items of indicators may vary depending on whether the spatial area of the city is a city unit or project unit, the development of spatial units of diagnostic and evaluation indicators should also be considered in the future. In addition, the goals of each plan are different in the current situation, and there is no integrated plan that considers sustainability and

smartness together. Therefore, it is necessary to improve the indicator system considering the type of plan.

To develop into a smart sustainable city, it is necessary to establish frameworks and indicators through smart sustainability initiatives, develop planning models, and implement pilot projects based on them. Moreover, securing completeness of the policy through a review of the appropriateness of initiative strategies is required. In addition, this process requires efforts to reflect the framework and indicators of smart sustainable cities by spatial hierarchy. This could be achieved by expanding the scope and application of integrated land-environmental planning management, which has been promoted in Republic of Korea since 2018.

The limitations and implications of the diagnostic indicators developed in this study are as follows: First, it concerns the spatial scope of the evaluation target plans. As evaluating plans for cities and developmental levels using the same indicators, there are limits to the ease of evaluation as it is established based on the plan structure and indicator protocols. So, it is necessary to adjust the protocol of indicators according to the plans. Second, it concerns differences in established purposes via the plan. In this study, Urban Master plans, Comprehensive Plans for Smart Cities and smart-city-type Urban Regeneration Revitalization Plan have different establishment purposes, but the same system indicators are applied to each plan. There is a limit to the lack of indicator consideration that can occur in achieving the plan objectives. In future research, it is necessary to improve the evaluation system so that it can select strategic indicators while considering goal and objective conformance, such as through a screening process, optional indicators, and weight setting. Last, it concerns smartness and sustainable development reflection levels via the plan. The Act on the Promotion of Smart City Development and Industry [48] was enacted in 2017 in Republic of Korea. For plans established before then, the smart city concept was limitedly applied. The level of smartness varies depending on when the plan was established. After the agreement at the end of 2015, the SDGs became a global issue, and the concept of sustainable development was only partially applied. Therefore, when establishing plans, the evaluation indicators and implications of the results should be considered via re-evaluation, monitoring, and feedback systems.

*5.2. Discussion*

The necessity and justification of development into smart sustainable cities are expected to continuously increase in the future. This is expected to further accelerate due to the rapid development of digital technology and increased frequency of impacts and damages caused by climate change and disaster. Despite the need for urban development, there are still challenges to be solved in planning and implementation.

The first challenge is information exposure and damage to individuals and institutions in the process of producing and utilizing information used for urban planning and management. To minimize this impact, it is necessary to secure cyber resilience, including the advancement of block chain technology. In smart sustainable cities, various information, such as sensor-based information to collect environmental, social, and economic information, information produced by citizens' movement and socio-economic activities, and information produced by citizens based on living lab, are produced. In the process of producing and utilizing such information, if individuals, households, and institutions are not protected, physical, economic and mental damage can occur. Therefore, it is necessary to build trust so that economic entities are protected.

Second, the application and use of smart technology should be appropriately introduced based on the goals of urban development, regional conditions, and social acceptance. The introduction of technologies that meet the city's SDGs should be prioritized, rather than focusing on competition with other cities for introducing the best technology. This should be based on the regional characteristic and understanding of citizens in accepting technology. As shown in some smart cities in cases of other countries, efforts to secure residents' acceptance are important, as residents' opposition would force the project to stop.

In this respect, acceptance of the urban technology is one of the key issues in sustainable urban technology along with the trust in the executor for information protection.

Third, smart cities are currently being promoted for new cities or areas with a good urban base; however, a smart sustainable city plan is likely to be more necessary for existing cities. Smart urban revitalization requires securing the sustainability of the declining area; however, as it is promoted by demand and participation, smart technology is being introduced at a limited level due to differences in technological development and understanding. For a more innovative approach, matching technologies suitable for the problems of existing cities should be appropriately introduced, and urban revitalization should be promoted by combining them. In other words, it is necessary to discover hardware and software technologies suitable for the level of development of new and existing cities, and various models that synthesize them are required. The discovery and promotion of the smart green city model project currently being executed by the government can contribute to future development.

Fourth, it is necessary to discover the overlapping area of sustainability and smartness. The sustainability of smart cities is in-line with the development of smart cities in terms of expanding the criticism of the technology bias of existing technology-based digital smartness. Therefore, smart sustainable cities need to embody components in the intersection of low-carbon green cities considering global climate change and changes in the energy supply system and smart cities based on ICTs technology. In order to build an area where sustainability and smartness intersect, we need to consider components of sustainability such as Low-carbon Green City, NetZero Energy City, Resilient City, and Citizen Participation-led Living Lab and components of smartness such as Service Technology Infrastructure and Management (STIM) in an integrated manner [81,82].

Smart sustainable cities are the direction and future of cities. This study is meaningful in that it derives major issues to be considered in these aspects and suggests tasks to be improved in the future. First, this study suggests directions for future research in aspects of smart technology discovery. Smart technologies and planning techniques for each sector that can be used to achieve sustainability must be discovered. Second, it is about the reflection of evaluation items and planning guidelines. The parts in the existing evaluation and guidelines scored high, but items without such reflections were excluded. Institutional improvement is required to use evaluation indicators as plan guidelines. Third, it concerns suggestions for planning guidelines. The overall low evaluation score is because the required contents are mentioned, but their specificity and appropriateness are insufficient. There is a need for guidelines on plan contents to supplement the insufficient parts. In addition, we need to consider the evaluation indicator applications. The evaluation indicators should consider plan and spatial unit, characteristics when classifying common and optional indicators. Last, future research should examine integrated planning models for smart sustainable cities, current issues and future-oriented planning models, the expansion of smart sustainable cities into spatial evaluation systems, and sustainable innovative technologies and support systems.

**Author Contributions:** Conceptualization, H.-S.C.; methodology, H.-S.C.; software, S.-K.S.; validation, H.-S.C.; formal analysis, H.-S.C. and S.-K.S.; investigation, H.-S.C. and S.-K.S.; resources, H.-S.C. and S.-K.S.; data curation, H.-S.C. and S.-K.S.; writing—original draft preparation, H.-S.C. and S.-K.S.; writing—review and editing, H.-S.C. and S.-K.S.; visualization, H.-S.C. and S.-K.S.; supervision, H.-S.C.; project administration, H.-S.C.; All authors have read and agreed to the published version of the manuscript.

**Funding:** This research was funded by Korea Environment Institute (WO2020-20) and Korean Ministry of Land, Infrastructure and Transport (22TSRD-B151228-04).

**Institutional Review Board Statement:** Not applicable.

**Informed Consent Statement:** Not applicable.

**Data Availability Statement:** Not applicable.

**Acknowledgments:** This paper was written following the research work "Direction for a Transition toward Smart Sustainable Cities based on the Diagnosis of the Smart City Plans (WO2020-20)" funded by the Korea Environment Institute (KEI)", and "Urban Declining Area Regenerative Capacity-Enhancing Technology Research Program" which was conducted by KEI (2022-025(R)) and funded by Ministry of Land, Infrastructure and Transport of Korean government (22TSRD-B151228-04).

**Conflicts of Interest:** The authors declare no conflict of interest. The funders had no role in the design of the study; in the collection, analyses, or interpretation of data; in the writing of the manuscript; or in the decision to publish the results.

## Appendix A

**Table A1.** Indicator review.

| Indicator | | Purpose and Assessment | Implications |
|---|---|---|---|
| **ISO** | 37120 (Quality of Service and Life in the City) | Focusing on urban services and quality of life and developing sustainability as principle. Consists of 19 categories, 104 indicators in total. | Smartness and resilience are key to urban sustainability. Covers most of urban components (environment, education, safety, leisure, transportation, sewage, water resources, finance, etc.). |
| | 37122 (Smart City) | Measurement of transition to smart city. Consists of a total of 85 indicators in the same category as ISO 37120. | |
| | 37123 (Resilient City) | Measurement of Urban Resilience from environmental, Social, and Economic Impact and Stress. Same category as ISO 37120. | |
| ETSI (Key Performance Indicators for Sustainable Digital Multiservice Cities) | | Evaluate Smart Cities and Smart City Projects separately. Sub-items (76 detailed indicators) classified into people, regional environment, economic growth, governance, and distribution/expansion. | Based on expert evaluation, results are provided through the City Tune service that supports the decision-making process of local governments and administrators for smart city development. The overall direction and framework are the same; however, the indicators are divided into the smart city itself and smart city projects (projects of various spatial hierarchies to make transition to smart city). |
| ITU | 4901 | Focus on ICT application. Criteria: ICT, environmental sustainability, productivity, quality of life, equity and social inclusion, physical infrastructure (48 indicators). | Efforts shall be made not to lose the uniqueness of cities (history, culture, economy, society, politics, etc.) due to the application of uniform evaluation indicators by establishing performance indicators reflecting the characteristics of each region and continuously modifying the evaluation method of performance indicators. |
| | 4902 | Focus on impact of sustainability. Criteria: environmental sustainability, productivity, quality of life, equity and social inclusion, physical infrastructure (30 indicators). | |
| | 4903 | Focus on Evaluating SDGs. Criteria: economy, environment, society, and culture (52 indicators). | |
| SDG11 | | It is a "sustainable city and residence" that contains contents on the creation of inclusive, safe, resilient, and sustainable cities and residences. | It is reflected based on the evaluation and standardization index of smart cities abroad. |
| APA, Sustainability Place Assessment | | Developed to evaluate the sustainable development of comprehensive urban planning. Evaluate 10 items (85 detailed items). | It has more solid evaluation framework than other urban plan evaluation indicators. |
| Roland Berger, Smart City Strategy Indicators | | Development of smart city strategic indicators to measure the city's comprehensiveness and goals for the core elements of smart city. The criteria are largely divided into planning, infrastructure/policies, and implementation, and a total of 31 detailed indicators are used. | It focuses on smart technology more than any other cases and must be considered from a technical perspective. Need to consider smart technology along with sustainability. |
| K-SDG11 | | It corresponds to the Sustainable City and Residence indicators among Korea's SDGs. | As various stakeholders participate and organize indicators, they reflect detailed indicators considering national characteristics unlike global indicators. |

**Table A2.** Indicator review.

| Indicator | Purpose and Assessment | Implications |
|---|---|---|
| KRIHS | As an indicator for evaluating the degree of maturity of smart cities, weights are derived through the evaluation of four existing cities, and a total of 150 final indicators and final guidelines are presented. | Compose evaluation framework by classifying target and means diagnostic indicators. |
| KICT | Classified into ICT, environmental sustainability, productivity, quality of life, equity and social integration, and physical infrastructure. | Evaluation index for the discovery of commercialization model that can lead the smart city. Consideration of service purpose, demand unit service, and element technology. |
| Ubiquitous Urban Planning | It is a guideline for establishing smart city plans. | Establish comprehensive planning to respond flexibly to condition changes in the future, such as the possibility of applying new technologies. |
| Guidelines for Urban Revitalization linked to SDG | Guidelines for urban revitalization linked to SDGs to promote environmental, economic, and social aspects, which are the main pillars of sustainable development, in order for urban regeneration projects to be sustainable. | Suggest planning elements to reflect sustainability in consideration of urban revitalization projects and characteristics of target sites. |

Source: [33–39,52,64–72].

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
