# Peer review of "Direction for a Transition toward Smart Sustainable Cities based on the Diagnosis of Smart City Plans"

_smartcities, doi:10.3390/smartcities6010009_

Round 1

Reviewer 1 Report

The paper aims to evaluate the planning process of sustainable urban systems. It is based on an interesting idea and certainly presents a relevant topic. However, it lacks the structure of an academic paper, it has several major drawbacks and its originality is yet to be determined. Below is a list of remarks:

1)      Structure. The paper would benefit from a rearrangement in order to ensure that it has a structure of an academic research paper which would, in general, follow a typical (Introduction)-(Method)-(Case study analysis)-(Results and discussion)-(Conclusion) framework.

2)      Introduction. A typical introduction should include problem context, literature review of similar studies and the hypothesis based on the gap analysis of the previously published research. As the literature

a.       Brief content. The introduction segment should be enhanced by adding a final paragraph that shortly highlights the key segments of the paper.

3)      Literature review. The authors did provide a number of references, but a more thorough literature review should be conducted and the specific science gap outlined. Emphasis should be on a targeted literature review that will provide justification for the research that your paper is going to cover.

4)      Method. Developing a suitable methodology is crucial. The authors did not describe their research principles rendering their results completely non-replicable. In addition, in present form, the lack of a novel, coherent method highlights the question of originality. What is the original, scientific contribution of the paper? In present form, the paper lacks an original academic contribution.

5)      Data. The paper has done substantial quantitative work. However, data employed in the study has not been clearly elaborated. Authors should invest more time in validating and describing the data use, as well as the sources of data. This is a major point, so please revise the paper accordingly.

6)      Quality of presentation. The quality of presentation is below acceptable. Some figures (i.e. 1, 2) should be made more attractive, some are impossible to read (which is a major issue) and some (copied from other literature) should perhaps be avoided. In addition, Table 1 should be improved to offer a more intuitive way for readers to understand its meaning and should be moved in the appendix, as in current size and form disrupts the flow of text in the manuscript.

7)      Discussion. This segment should be enhanced with additional outcomes and comparisons with other studies in the field. Are your results aligned with those of other studies in the field? Also, the benefits of the proposed model are not clear enough and should be described more concisely.

8)      Conclusion. General discussion in the final chapter should be limited, as there is no need for repeating well-known issues. On the other hand, practical implications of your research should be elaborated in more detail. It would be meaningful if you would discuss the results by suggesting how they are relevant to further sector development and how your research could be used in practice.

a.       Another important note is to avoid drawing conclusions which do not directly emanate for the research itself.

9)      Originality. Once again, I will emphasise the need for the authors to outline the originality of their research. Why does your research result in an expansion of knowledge on the subject relevant enough to be published by the Journal? In present form, that does not seem to be the case.

Author Response

Thank you for your helpful comments and review of our manuscript. Your insights led to improvements in our study. Please review the revised manuscript.

Answer to comment 1

We have rearranged the structure of the paper to reflect your comments.

Answer to comment 2

We have incorporated your comments into the literature review section, rather than the introduction, to highlight the background and necessity of our study.

The literature review section includes the need for this study based on the context of the problem, a review of similar studies, and the gap analysis with previously published studies.

We have also added a paragraph briefly introducing the key segments of the study.

Answer to comment 3

Our target literature includes previous studies that evaluate sustainable and smart cities. It was challenging to include all of these studies in the manuscript; therefore, we have attached them as a table in the appendix. According to your suggestion, we have added a paragraph regarding the improvements we have found in these studies

Answer to comments 4, 5 and 9.

Through the literature review, we found that existing smart cities and evaluation systems guiding them in a desirable direction does not sufficiently imply sustainability. Accordingly, we created a framework and indicators to evaluate the sustainability and smartness of urban planning comprehensively. There are concerns that quantitative indicators such as smart home ratio and smart city budget may fail to reflect regional characteristics and foster uniform plans. Therefore, we created a system that allows local experts to rate plans on a scale of 1–3 using content analysis. To avoid the establishment of a uniform plan that does not reflect regional characteristics, our indicators only present areas to be considered in planning for sustainability and smartness and some examples of technologies; essential technical elements are not mentioned in them. Experts who comprehensively considered the current status of the region, the plan's goals, and its vision were asked to judge whether the contents of the plan were appropriate and whether the goals could be achieved. Although being easy to use in a relatively simple manner, content analysis lacked data reliability (objectivity); therefore,, this study presented clear scoring criteria for evaluation objectivity. As such, the results would be the same if the same evaluators were to conduct the same evaluation again. In addition to increase objectivity, we created a discussion process among experts during the evaluation process and adjusted their scores.

This planned evaluation through indicators presents a desirable vision of the future and evaluates the quality of planning to see if it can be achieved, thereby enabling objective comparison between cities and providing a guide for setting directions and improving plans.

Answer to comment 6

As you have suggested, we have improved the quality of the figures and edited the text. In response to your comments, we have edited the table to improve the readability.

However, Table 1 is one of the main outputs of our paper, and so we believe it would be better to include it in the main manuscript rather than in the appendix.

Answer to comments 7 and 8.

We have systematically summarized the main results and supplemented the conclusion by adding imitations and implications as per your comments. In addition, discussions were supplemented by including studies that will be needed in the future.

Thank you very much for your attention, and we hope that our revised manuscript will be acceptable for publication in Smart Cities.

Reviewer 2 Report

The article concerns the interesting topic of study on "Direction for a Transition toward Smart Sustainable Cities based on the Diagnosis of the Smart City Plans". The subject of the article is current and in line with the thematic scope of the journal. Moreover, the subject matter relates to the improvement of the current state of sustainable development.

Overall, the article is very interesting and in my opinion a valuable contribution to the emerging literature on sustainable development and smart cities. However, I have a few suggestions that could improve the quality of the article.

From an editing point of view:

1. Please correct Figure 2.

2. Please justify the text.

3. Figure 4 is very illegible - improve its quality.

4. Figure 7 is also illegible.

From the substantive point of view:

1. In the introduction, please provide a paragraph presenting the structure of the article, explaining what the individual sections present.

3. In chapter 2.2. please refer more broadly to transport issues in relation to smart cities, especially to new mobility services such as vehicle sharing.

Check and refer to: 10.3390/joitmc8010036, 10.9770/jesi.2021.9.1(1), https://doi.org/10.3390/smartcities5030054.

4. The discussion should be separated from the summary.

5. In the discussion, refer to research conducted by other scientists on the same topic.

6. In the summary, indicate whether the goal of the work has been achieved. Add recommendations or conclusions. The summary is too short, it should also be extended with further research plans of the authors and research limitations.

Good luck!

Author Response

Thank you for your helpful comments and review of our manuscript. Your insights led to improvements in our study. Please review the revised manuscript.

Answers to comments 1-4 from an editing point of view

We improved the quality of the figures and edited the text.

Answer to comment 1 from the substantive point of view

As you suggested, we have added a paragraph to the introduction to explain the individual sections.

Answer to comment 3 from the substantive point of view

We reviewed the literatures you suggested and reflected them in our paper.

Our evaluation indicators deal with the transportation sector as a sustainable smart city strategy, which can evaluate whether vehicle sharing is appropriately reflected in plans.

Answer to comment 4 from the substantive point of view

As you suggested, we separated the summary from the discussion.

Answer to comments 5-6 from the substantive point of view

As you have suggested, we have systematically summarized the main results and supplemented the conclusion by adding imitations and implications. In addition, discussions were supplemented by including studies that will be needed in the future.

Thank you very much for your attention, and we hope that our revised manuscript will be acceptable for publication in Smart Cities.

Reviewer 3 Report

Dear authors,

Thank you for sending this paper to the journal Smart Cities. It's an interesting work. There are my comments:

Comment 1: Part 1 and Part 2 - more literature review for the diagnosis of the smart city plans, such as mobile signal data, social sensing data, and machine learning related to decision-making researches. Perhaps some recent refs are, for example: 1) Personal Information Protection and Interest Balance Based on Rational Expectation in the Era of Big Data: A Case on the Sharing of Mobile Phone Signaling Big Data in Smart City Planning, International Review for Spatial Planning and Sustainable Development 2022, 10(1):1-23; 2) Does social perception data express the spatio-temporal pattern of perceived urban noise? A case study based on 3,137 noise complaints in Fuzhou, China, Applied Acoustics 2022, 201:109129.

Comment 2: Figure 2 - Perhaps a consideration of your research about how to apply your results of evaluation for government and platform system.

Comment 3: Smart sustainable city framework looks interesting, but the Figure is very vague. Also, perhaps smart enables includes automation, and city responsibility includes commuter traffic.

Comment 4: I wonder detailed spatial form indicators for city evaluation.

Comment 5: Figure 6- The units of the vertical axis are missing.

Comment 6: Almost no data in interval [2, 3]. Plesase consider the interval of the axes to make the result easier to understand

Comment 7: Limitation should be mentioned.

Best regards

Author Response

Thank you for your helpful comments and review of our manuscript. Your insights led to improvements in our study. Please review the revised manuscript.

Answer to comment 1

We reviewed the papers you suggested and reflected them in our paper.

Our evaluation indicators consider the areas you propose to be key issues in sustainable smart city strategies and implementation. For example, if an action plan for personal information protection is properly established in the implementation sector, experts assign a high score to that indicator when evaluating the plan.

Answer to comment 2

Figure 2 shows a schematic diagram of the study, modified, and moved to the introduction session.

Answer to comment 3

The framework has improved quality so you can see better.

Answer to comment 4

Because our evaluation system was based on content analysis, quantitative spatial indicators were not used. There are concerns that quantitative indicators such as smart home ratio and smart city budget may fail to reflect regional characteristics and foster uniform plans. Therefore, we created a system that allows local experts to rate plans on a scale of 1–3 using content analysis. To avoid the establishment of a uniform plan that does not reflect regional characteristics, our indicators only present areas to be considered in planning for sustainability and smartness and some examples of technologies; essential technical elements are not mentioned in them. Experts who comprehensively considered the current status of the region, the plan's goals, and its vision were asked to judge whether the contents of the plan were appropriate and whether the goals could be achieved. Although being easy to use in a relatively simple manner, content analysis lacked data reliability (objectivity); therefore,, this study presented clear scoring criteria for evaluation objectivity. As such, the results would be the same if the same evaluators were to conduct the same evaluation again. In addition to increase objectivity, we created a discussion process among experts during the evaluation process and adjusted their scores.

Answer to comment 5

We have modified figure 6 to reflect your comments.

Answer to comment 6

As shown in Figure 4, there were criteria for a score of 0–3. However, it was still difficult to get 2-3 points. However, we think that the fact that there are no two or three points is very meaningful in that we can identify the areas that need to be improved in future plans.

Answer to comment 7

As suggested, we have supplemented the conclusion by adding imitations and implications

Thank you very much for your attention, and we hope that our revised manuscript will be acceptable for publication in Smart Cities.

Round 2

Reviewer 1 Report

The authors revised the paper according to the remarks made.

Reviewer 2 Report

Tha article was improved and is ready for publication.

Reviewer 3 Report

My comments have been clarified in this Revision.